# REmoNet: Reducing Emotional Label Noise via Multi-regularized Self-supervision

Wei-Bang Jiang*
Shanghai Jiao Tong University
Shanghai, China
935963004@sjtu.edu.cn

Yu-Ting Lan*
Shanghai Jiao Tong University
Shanghai, China
lanyuting8806@gmail.com

Bao-Liang Lu†
Shanghai Jiao Tong University
Shanghai, China
bllu@sjtu.edu.cn

## ABSTRACT

Emotion recognition based on electroencephalogram (EEG) has garnered increasing attention in recent years due to non-invasiveness and high reliability of EEG measurements. Despite the promising performance achieved by numerous existing methods, several challenges persist. Firstly, there is the challenge of emotional label noise, stemming from the assumption that emotions remain consistently evoked and stable throughout the entirety of video observation. Such an assumption proves difficult to uphold in practical experimental settings, leading to discrepancies between EEG signals and anticipated emotional states. In addition, there's a need for comprehensive capture of temporal-spatial-spectral characteristics of EEG signals and cope with low signal-to-noise ratio (SNR) issues. To tackle these challenges, we propose a comprehensive pipeline named REmoNet, which leverages novel self-supervised techniques and multi-regularized co-learning. Two self-supervised methods, including masked channel modeling via temporal-spectral transformation and emotion contrastive learning, are introduced to facilitate the comprehensive understanding and extraction of emotion-relevant EEG representations during pre-training. Additionally, fine-tuning with multi-regularized co-learning exploits feature-dependent information through intrinsic similarity, resulting in mitigating emotional label noise. Experimental evaluations on two public datasets demonstrate that our proposed approach, REmoNet, surpasses existing state-of-the-art methods, showcasing its effectiveness in simultaneously addressing raw EEG signals and noisy emotional labels.

## CCS CONCEPTS

• **Human-centered computing** → HCI design and evaluation methods; • **Computing methodologies** → **Artificial intelligence**; **Cognitive science**.

## KEYWORDS

EEG, emotion recognition, self-supervised learning, noisy labels

---

*Both authors contributed equally to this research.
†Corresponding author

**ACM Reference Format:**
Wei-Bang Jiang, Yu-Ting Lan, and Bao-Liang Lu. 2024. REmoNet: Reducing Emotional Label Noise via Multi-regularized Self-supervision. In *Proceedings of the 32nd ACM International Conference on Multimedia (MM '24), October 28-November 1, 2024, Melbourne, VIC, Australia*. ACM, New York, NY, USA, 10 pages. https://doi.org/10.1145/3664647.3681406

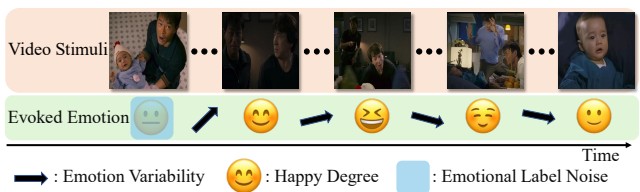

**Figure 1: Illustration of emotion evoked progression. Specifically, as the video progresses, the happy degree of subjects gradually amplifies, peaking at the moment of laughter. Following the laughter peak, the intensity gradually diminishes over time.**

## 1 INTRODUCTION

Understanding human emotions is a fundamental aspect of human-computer interaction, affective computing, and psychology research [20]. To achieve this, various modalities have been utilized including non-physiological signals like facial expression [23, 39] and speech [2, 37], physiological signals like electroencephalography (EEG) [14, 19] and electrocardiogram [13, 33]. Among diverse modalities, EEG has garnered significant attention due to its non-invasive nature, high temporal resolution, and direct correlation with neural activity. One of the mainstream approaches to studying human emotions with EEG signals involves classifying predefined emotion categories evoked by video stimuli using computational methods, such as deep neural networks [42]. Notable benchmarks include the SEED series [47, 48], which comprises EEG recordings from multiple subjects across multiple sessions in response to carefully selected videos. Researchers have explored various powerful neural network architectures within these benchmarks, leading to significant advancements in EEG-based emotion recognition [18, 44].

However, current methods still overlook the complexity of such paradigms for evoking emotions, which poses significant challenges. The first challenge is emotional label noise, due to the assumption within these paradigms that emotions remain evoked and stable throughout the entire duration of video observation. Such assumption is inherently challenging to achieve in practical experimental settings, often resulting in discrepancies between EEG signals and

the anticipated evoked emotional states. Secondly, the comprehensive capture of temporal-spatial-spectral characteristics of EEG signals is another hard nut to crack. This is because comprehensively understanding and interpreting these multifaceted characteristics present challenges in the realm of emotion recognition research. Thirdly, low signal-to-noise ratio (SNR) remains a tough issue. The acquisition of EEG signals is susceptible to interference from biological and environmental noise, impeding the attainment of precise EEG recordings and hampers the ability to precisely capture informative emotion-relevant EEG representations.

Figure 1 illustrates the process of emotion-evoking and further explains these challenges. When a subject is exposed to visual stimuli of videos, brain responses exhibit continuous and dynamic fluctuations [8, 16]. For instance, in response to a happy video stimulus, subjects' pleasure gradually increases, peaking at the laughter point, before gradually diminishing. This process of acclimatization contributes to inaccuracies in emotion labeling for applying identical labels to all data derived from the same videos and introduces emotional label noise. Moreover, the intricate and subtle variations in emotions necessitate the neural networks to comprehensively understand and interpret the multifaceted characteristics of EEG signals and discern the most emotion-relevant factors.

To address the aforementioned challenges, we introduce a comprehensive pipeline REMoNET, namely, **R**educing **Emo**tional label noise **Net**work via Multi-regularized self-supervision. We propose two novel self-supervision methods to comprehensively understand and interpret the multifaceted characteristics of EEG signals and extract the most emotionally relevant representations. Firstly, we propose masked channel modeling via temporal-spectral transformations (MCM-TST), which is inspired by the insight to capture the distinct frequency bands associated with different brain activities, such as alpha waves representing relaxation and beta waves indicating heightened mental activity. Secondly, we present emotion contrastive Learning (ECL), where true-negative pairs are selected based on the properties of video stimuli trials, facilitating the extraction of emotion-relevant representations. To combat emotional label noise, we propose a novel approach called multi-regularized co-learning. This method requires two different neural networks that are pre-trained by MCM-TST and ECL, and leverages their ability to provide two different views of data, guiding the training process by exploiting the feature-dependent information through the intrinsic similarity and avoiding bias caused by noisy labels. Our innovative multi-regularization strategy enhances the diversity of the two networks, gradually aligning them in agreement with each other, albeit through different manners, and resulting in significant performance improvements. We conduct comprehensive experiments on two public datasets, SEED [48] and SEED-IV [47], to evaluate the effectiveness of REMoNET. Experimental results demonstrate that our method achieves SOTA performance among both self-supervised learning and supervised learning methods.

## 2 RELATED WORK

### 2.1 EEG-based Emotion Recognition

EEG-based emotion recognition has witnessed significant advancements in recent years, particularly with the emergence of deep learning techniques. Most of the existing works rely on hand-crafted

features due to their ability to capture the spectrum inductive bias of brain signals. These features, including power spectral density [7], differential entropy (DE) [6], and differential asymmetry [29], have been extensively investigated. To represent the spatial and temporal dimensions inherent in EEG data, various architectures have been proposed [15, 17, 25, 27, 46, 49].

Although studies using hand-crafted features are thriving, such complex feature extraction is usually slow and of low-efficiency, as well as needs prior knowledge. There are also a few end-to-end studies focusing on raw EEG signals. Schirrmeister *et al.* proposed deep and shallow convolutional neural networks to process raw EEG signals by two-stage spatial and temporal convolution [34]. Larwhern *et al.* introduced the depth-wise and separable convolutions and proposed EEGNet to extract temporal and spatial information from EEG [22]. Recently, Ding *et al.* exploited a multi-scale convolutional neural network called TSception which learns discriminative representations in the time and channel dimensions simultaneously [5]. However, end-to-end approaches face the problem that it's difficult for models to understand and interpret the multifaceted characteristics of EEG signals and extract the most emotionally relevant representations, which leads to relatively poor performance.

### 2.2 Self-supervised Learning

Self-supervised learning, which is arising in the last few years, focuses on learning generic representations for downstream tasks from massive unlabeled data. It can be roughly categorized into two groups, i.e., contrastive learning and masked signal modeling.

Contrastive learning aims to learn discriminative representations to distinguish a sample from others. It predefines positive and negative sample pairs, and trains the model to minimize (maximize) the distances between positive (negative) pairs. There are some influential works, e.g., MoCo [11], SimCLR [3]. Mohsenvand *et al.* extended the SimCLR framework to EEG signals [32]. Shen *et al.* proposed a contrastive learning method for inter-subject alignment to perform cross-subject emotion recognition [35].

Masked signal modeling typically masks a proportion of signals and reconstructs the masked parts from unmasked signals, and it focuses on learning local structural relations within a sample. He *et al.* presented the Masked Autoencoder (MAE) to reconstruct the raw pixels from partially observed image patches [10]. MaskFeat [40] introduced the low-level hand-crafted features as the reconstruction targets to learn more semantic features. [26, 44] extended MAE for EEG-based emotion recognition, and they randomly masked some channels of the DE features and reconstructed them. However, they are not in an end-to-end manner.

### 2.3 Learning with Noisy Labels

There are various studies on robust learning with noisy labels. One group is sample selection. Co-teaching series [9, 45] maintain two networks and select small-loss samples to update the peer network. JoCoR [41] introduced co-regularization to reduce the diversity of two networks. However, these methods need prior knowledge such as noise rates and data distributions. Another group is learning with self-supervision. Tan *et al.* practiced co-learning which imposes constraints of structural similarity with self-supervised learning to cope with noisy labels [36]. Inspired by these works, we

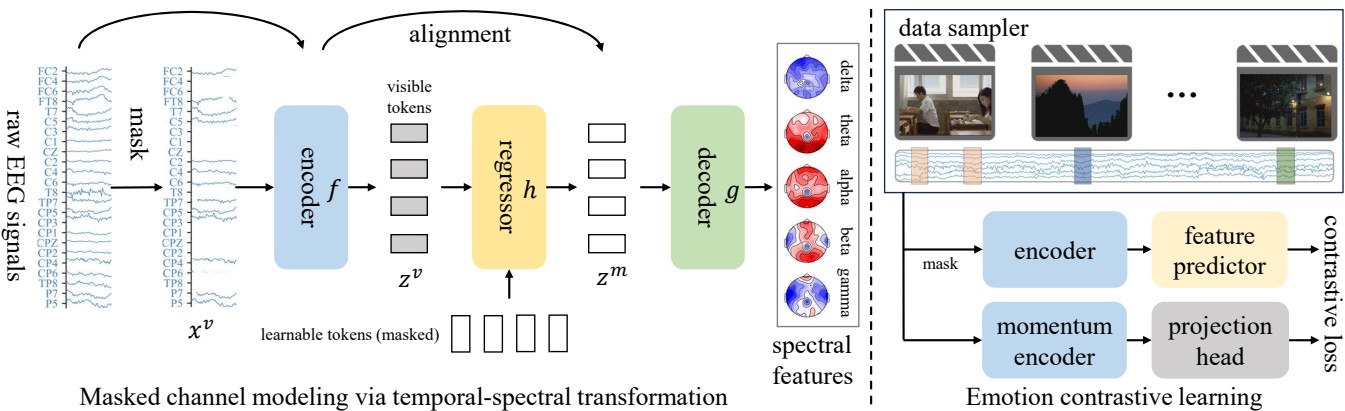

**Figure 2: Illustration of the pre-training stage. Left: MCM-TST. We mask a portion of EEG channels and predict the spectral features from the unmasked parts. Right: ECL. We sample the positive and negative samples from the data sampler and update the network by exponential moving average. In the data sampler, EEG segments with the same color denote positive pairs while the ones with different colors denote negative pairs.**

further present distance-wise and angle-wise regularization with self-supervised learning to combat noisy emotion labels. Li *et al.* [24] manually made noisy emotion labels and proposed a capsule network using a joint optimization strategy. To the best of our knowledge, there are few studies addressing noisy emotional labels in end-to-end EEG-based emotion recognition.

## 3 METHODOLOGY

### 3.1 Preliminaries

We formulate the EEG signals as $X \in \mathbb{R}^{C \times T}$, where $C$ is the number of EEG channels, and $T$ is the number of time steps. $X$ is then sliced into $X = (x_1, x_2, ..., x_N)$ by the time window $T_w$ with non-overlap, where $x_i$ follows the distribution $\mathcal{X} \subset \mathbb{R}^{C \times T_w}$ and $N = \frac{T}{T_w}$. We denote the labels as $Y = (y_1, y_2, ..., y_N)$, where $y_i$ follows the distribution $\mathcal{Y} \subset \{0, 1\}^c$ and $c$ is the number of classes. Note that $Y$ may follow a noisy joint distribution with $X$. The objective is to find a mapping function $\mathcal{F} : \mathcal{X} \rightarrow \mathcal{Y}$ that minimizes the empirical risk $\mathcal{R}_{\mathcal{L}}(\mathcal{F})$ under the loss function $\mathcal{L}$:

$$\mathcal{R}_{\mathcal{L}}(\mathcal{F}) = \frac{1}{N} \sum_{i=1}^{N} \mathcal{L}(\mathcal{F}(x_i), y_i). \quad (1)$$

### 3.2 Pre-training with Self-supervision

The self-supervised pre-training stage enables the model to learn more general and emotion-related representations from unlabeled EEG data. As shown in Figure 2, we adopt two kinds of self-supervised learning methods, i.e., masked channel modeling temporal-spectral transformation and emotion contrastive learning, as pretext tasks. Considering their respective properties, the former will be used for subsequent fine-tuning to predict emotions while the latter will serve as regularization to counteract noisy labels. Since the pre-training benefits from a large amount of data, we use the training data of all subjects as the training set.

*3.2.1 Masked Channel Modeling via Temporal-Spectral Transformation.* To comprehensively understand and interpret the multifaceted

characteristics of EEG signals and extract the most emotion-relevant representations, we propose MCM-TST. The motivation is that spectral features, especially with high-frequency bands, contain more semantic and emotional information for decoding human emotions, which is more beneficial for the model pre-training than reconstructing the semantically sparse and noisy original EEG signals.

To make the model learn more effectively and efficiently [4], the pre-training of masked EEG modeling has three components, encoder $f$, regressor $h$, and decoder $g$. For each input EEG sample $x$, we randomly mask a ratio of $r$ channels, resulting in $x^m$ for masked channels and $x^v$ for visible channels. The encoder $f$, whose architecture will be described in detail later, maps the visible channels $x^v$ into latent representations $z^v$. It only processes visible EEG signals and aims at extracting features for reconstructing representations of masked EEG in a latent space.

The regressor $h$ is designed to predict the latent representations $z^m$ for masked EEG channels $x^m$ from visible EEG channels $x^v$. It comprises $L_h$ Transformer blocks [38] with cross-attention. The queries $Q^m$ which have the same number of tokens as $x^m$ are set learnable and optimized by the whole network during training. The keys and values are calculated from $x^v$ which is the output of the encoder $f$ and contains information about the entire EEG sample. To encourage the model to encode EEG rationally in the latent space, an alignment constraint is imposed on the latent representations $z^m$. The masked EEG channels $x^m$ are passed to the encoder $f$, therefore, generating $\hat{z}^m$. Then, $z^m$ is aligned with $\hat{z}^m$ by the alignment loss

$$\mathcal{L}_{align} = \frac{1}{N} \sum_{i=1}^{N} \|\hat{z}_i^m - z_i^m\|_2^2. \quad (2)$$

As for the decoder $g$, it takes the latent representations of masked EEG channels $z^m$ as input and generates the corresponding objective spectral features $o^m$. It simply consists of $L_g$ Transformer blocks followed by a linear layer to match the target and it only handles the masked EEG channels to reduce the overhead. The learnable positional embeddings are added to $z^m$ before being fed into the decoder. We use the DE features as the objective features. The

mean-squared error (MSE) is used as the reconstructed loss

$$\mathcal{L}_{recon} = \frac{1}{N} \sum_{i=1}^{N} \|\hat{o}_i^m - o_i^m\|_2^2, \tag{3}$$

where $\hat{o}_i^m$ is the objective DE features. Specifically, the total loss for MCM-TST is

$$\mathcal{L}_{mfp} = \mathcal{L}_{recon} + \lambda \mathcal{L}_{align}, \tag{4}$$

where $\lambda$ is a trade-off parameter.

*3.2.2 Emotion Contrastive Learning.* The pre-training of ECL, as mentioned above, focuses on endowing the model with strong instance discriminability. As it has a better ability to learn structural similarity between instances, the model is pre-trained to play the role of regularization in the fine-tuning stage.

The primary problem in ECL is the definition of positive and negative samples. There are two different ways: data augmentation or data sampler. Data augmentation usually adopts methods such as additive noise, time shift, amplitude scale, etc. on original EEG signals to get augmented EEG data. Whereas, unlike image augmentation in computer vision, directly augmenting physiological signals like EEG is not appropriate to the characteristics of the signals. So, we choose a data sampler that samples the data from the training set. Based on the individual differences and the observation that samples from the same trials (video clips) are annotated with the same labels, we propose a simple but effective definition: only samples from the same video clips of the same subjects are regarded as positive pairs while others are negative pairs. Though different clips may be used to elicit the same emotion, they still differ in the impact of arousal or dominance. With this assumption, we hope the model has a strong discrimination ability with not only emotion categories but also arousal or dominance, which will be beneficial for subsequent fine-tuning or regularization.

We denote the input minibatch of data as $X_B$. We also randomly mask a portion of channels and get $X_B^m$. The masking operation not only accelerates the training process but provides a more robust pretext task where the model should learn global discriminative features from corrupted EEG signals [28]. The encoder $f$ is of the same architecture as the encoder in masked EEG feature prediction, and it transforms $X_B^m$ into $Z^m$ in latent space. Since $Z^m$ only contains information of partial EEG signals, we elaborate a feature predictor $m$ which predicts global representations $Z$ from masked features $Z^m$. The feature predictor $m$ is composed of $L_m$ Transformer blocks. The latent features $Z^m$ are first concatenated by learnable vectors that represent masked channels, then added by positional embeddings, and finally passed to $m$ to get the output $M$.

For the data sampler, it randomly selects one sample from each video clip of each subject in the training set, and the selected samples are denoted as $X_S$. Therefore, for each sample in the minibatch $X_B$, there are exactly one positive sample and $nk - 1$ negative samples in $X_S$ (assume that there are $n$ subjects and $k$ video clips in the training set for each subject). The EEG samples in $X_S$ are not masked and they are fed into a momentum encoder $f'$ which is a siamese network with the encoder $f$. The momentum encoder $f'$ is updated through exponential moving average (EMA) to obtain more stable representations for ECL:

$$\theta_{M'} \leftarrow \eta \theta_{M'} + (1 - \eta) \theta_M, \tag{5}$$

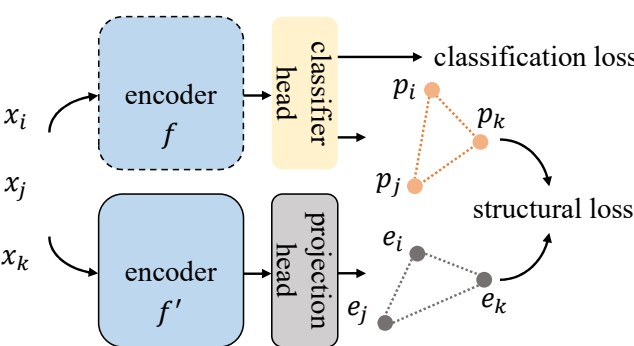

**Figure 3: The architecture of the multi-regularized fine-tuning. The solid line borders represent frozen parameters and the dashed line border means partially frozen parameters. The structural similarity loss is cooperated with the classification loss to cope with noisy labels.**

where $\theta_M$ and $\theta_{M'}$ are the parameters of the encoder $M$ and the momentum encoder $M'$. respectively, and $\eta$ is the momentum coefficient. There is a projection head $u$ consisting of a linear layer and a ReLU activation function that map the output of the momentum encoder $f'$ to $M'$ in the embedding space. The loss function for ECL is the InfoNCE loss:

$$\mathcal{L}_{contra} = -\sum_{m \in M} \log \frac{e^{m \cdot m'^+/\tau}}{e^{m \cdot m'^+/\tau} + \sum_{j=1}^{nk-1} e^{m \cdot m_j'^-/\tau}}, \tag{6}$$

where $m'^+$ denotes the positive sample in $M'$, $m'^-$ denotes the negative sample and $\tau$ is the temperature parameter.

## 3.3 Fine-tuning with Multi-regularized Co-learning

At the fine-tuning with multi-regularized co-learning stage, the generic model of each subject is fine-tuned independently to get the personalized model for emotion recognition. As mentioned before, noisy emotional labels at this stage can cause non-negligible degradation in performance. To overcome this problem, we impose multi-regularization through mutual relations of data samples with the help of the model pre-trained by ECL. There are two encoders $f$ and $f'$ as presented in Figure 3. The encoder $f$ is pre-trained by MCM-TST while the encoder $f'$ is pre-trained by ECL. The main task is supervised learning with classification loss and cross-entropy loss is used here:

$$\mathcal{L}_{ce} = -\sum_{i=1}^{N} y_i \log p_i, \tag{7}$$

where $p_i$ is the prediction and $y_i$ is the corresponding label.

For three input samples $x_i$, $x_j$, and $x_k$, we denote the output logits of the encoder $f$ followed by a classifier head $v$ as $p_i$, $p_j$, and $p_k$. The output features of the encoder $f'$ followed by the projection head $u$ are $e_i$, $e_j$, and $e_k$. The encoder $f'$ is assisted to maximize the agreement on the two encoders between the logits and features. To achieve this, we introduce structural similarity loss, which consists of distance-wise regularization and angle-wise regularization to prevent the model from overfitting on noisy labels. The multi-regularization implicitly handles noisy labels and does not need to

explicitly detect noisy labels. Therefore, it can be adaptively applied to any dataset with any noise level or even unnoisy labels, which can also improve the robustness and mitigate overfitting.

*3.3.1 Distance-wise Regularization.* The distance-wise regularization computes the distances between every two samples, minimizing the relative distances obtained by the two models [36]. We first calculate the Euclidean distance $d(p_i, p_j)$ and $d(e_i, e_j)$. It is not reasonable to directly minimize these two distances because they are not in the same feature space. Thus, we normalize the distances to $[0, 1]$ using a similarity metric $q$:

$$q(d) = e^{-\frac{1}{2}(\frac{d-\mu}{\sigma})^2}, \tag{8}$$

where $\mu$ is set to 0 and $\sigma$ is set to 0.5. The similarity metric $q$ satisfies that $\lim_{d\to\infty} q(d) = 0$ and $\lim_{d\to 0} q(d) = 1$. The distance-wise regularization is implemented by the Kullback-Leible divergence:

$$\mathcal{L}_{dis} = \sum_{i \neq j} q(d(e_i, e_j)) \log \frac{q(d(e_i, e_j))}{q(d(p_i, p_j))}. \tag{9}$$

*3.3.2 Angle-wise Regularization.* The angle-wise regularization is defined on triplet samples and it penalizes angular differences between the two encoders. The angle of every three samples is computed by angular metric $a$:

$$a(p_i, p_j, p_k) = \langle \frac{p_i - p_j}{\|p_i - p_j\|}, \frac{p_k - p_j}{\|p_k - p_j\|} \rangle, \tag{10}$$

where $\langle, \rangle$ represents the inner product. The angle-wise regularization loss is the smooth L1 loss on every triplet samples:

$$\mathcal{L}_{agl} = \sum_{i \neq j \neq k} l_\delta(a(p_i, p_j, p_k), a(e_i, e_j, e_k)), \tag{11}$$

where the smooth L1 loss $l_\delta$ is

$$l_\delta(x, y) = \begin{cases} \frac{1}{2}(x-y)^2, & \text{for} |x-y| \le 1; \\ |x-y| - \frac{1}{2}, & \text{otherwise.} \end{cases} \tag{12}$$

In all, the final loss is the linear combination of the cross entropy loss $\mathcal{L}_{ce}$, the distance-wise regularization loss $\mathcal{L}_{dis}$, and the angle-wise regularization loss $\mathcal{L}_{agl}$:

$$\mathcal{L}_{total} = \mathcal{L}_{ce} + \alpha \mathcal{L}_{dis} + \beta \mathcal{L}_{agl}, \tag{13}$$

where $\alpha$ and $\beta$ are the coefficients.

Note that the encoder $f'$ and the projection head $u$ are frozen during the whole fine-tuning stage. The encoder $f$ is partially frozen and it will be clarified later. The whole algorithm of the pre-training and fine-tuning stage is presented in Algorithm 1.

## 3.4 The Hybrid Architecture

We present the architecture of the encoder, as depicted in Figure 4. As the raw EEG signals have a high resolution in the temporal domain, it is critical to extract informative features and reduce dimension from the time series of EEG. The existing methods usually use 1-D convolution to process EEG signals on the temporal dimension. Here, we introduce the gated temporal convolutional networks (GTCN) due to their powerful ability in controlling information flow through layers [43]. For an EEG sample $x \in \mathbb{R}^{C \times T_w}$,

---

**Algorithm 1:** The process of pre-training and fine-tuning stages in REMoNet

**Input:** The pre-training EEG data $X$, the fine-tuning data with label $X_t$ and $Y_t$, the encoders $f$ and $f'$, the regressor $h$, the decoder $g$, the feature predictor $m$, the projection head $u$, the classifier head $v$.

**Output:** The trained encoder $f$ and the classifier head $v$.

**Pre-training Stage:**

1  Initialize the encoders $f$ and $f'$, the regressor $h$, the decoder $g$, the feature predictor $m$, the projection head $u$ ;
2  Mask the pre-training EEG data $X$ to get $X^v$ ;
3  Obtain the latent representations $Z^v = f(X^v)$, $Z^m = h(Z^v)$, and $\hat{Z}^m = f(X)$ to calculate the alignment loss $\mathcal{L}_{align}$ ;
4  Predict the masked spectral features $O = g(Z^m)$ and calculate the reconstructed loss $\mathcal{L}_{recon}$ ;
5  Optimize $f$, $h$, and $g$ by minimizing equation (4) ;
6  Compute $m(f'(X_B))$ and $u(f'(X_S))$ for ECL;
7  Optimize $f'$, $m$, and $u$ by minimizing equation (6) ;
   **Return:** $f$, $f'$, and $u$

**Fine-tuning Stage:**

8  Initialize the classifier head $v$ ;
9  Obtain the pre-trained $f$, $f'$, and $u$ ;
10 Calculate $P_t = v(f(X_t))$ and $E_t = u(f'(X_t))$ ;
11 Optimize $f$ and $v$ by minimizing equation (13) ;
   **Return:** $f$ and $v$

---

we apply two 1-D convolutions whose kernel sizes are equal to their strides. The gated mechanism can be formulated as

$$\tanh(\text{conv-a}(x)) \odot \sigma(\text{conv-b}(x)), \tag{14}$$

where $\odot$ denotes element-wise product and $\sigma$ denotes sigmoid activation function. Considering the masking operation in pre-training, the GTCN is applied only on the time dimension and there are no more convolutions across EEG channels. Batch normalization and GELU (Gaussian error linear units) activation [12] are followed by the gated operation. We stack $L_1$ GTCN blocks to get the low-dimensional features.

After the GTCN blocks, positional embeddings are added to the output features, which are initialized with sine and cosine functions and are learnable during training. Then, there are $L_2$ spatial Transformer blocks to learn the spatial features across EEG channels. We use the vanilla Transformer here, which is implemented by multi-head self-attention and feed-forward layer with skip connection and layer normalization. Overall, our encoder processes both the spatial and temporal domains of raw EEG signals simultaneously and extracts informative features for subsequent tasks.

## 4 EXPERIMENTS

### 4.1 Datasets and Preprocessing

We conduct comprehensive experiments on two public datasets SEED [48] and SEED-IV [47]. The most distinguishable difference is the number of emotions. SEED has 3 emotions (positive, neutral, and negative) while SEED-IV contains 4 emotions (happy, neutral,

| Method | SEED | | | | SEED-IV | | | |
|---|---|---|---|---|---|---|---|---|
| | ACC | STD | F1 | STD | ACC | STD | F1 | STD |
| SCN [34] | 70.71 | 13.88 | 68.90 | 15.75 | 42.34 | 13.13 | 40.70 | 12.90 |
| EEGNet [22] | 75.38 | 11.68 | 73.98 | 13.58 | 51.13 | 13.38 | 48.17 | 13.57 |
| FBCNet [31] | 72.76 | 12.88 | 71.38 | 14.63 | 50.72 | 08.89 | 45.64 | 11.28 |
| BENDR [21] | 67.69 | 12.04 | 66.51 | 13.26 | 45.94 | **08.46** | 44.90 | **09.50** |
| MAE [10] | 73.88 | **04.59** | 72.11 | **05.41** | 54.01 | 09.97 | 45.70 | 15.35 |
| TSception [5] | 76.58 | 11.92 | 75.20 | 13.68 | 55.29 | 13.42 | 51.87 | 16.33 |
| LaBraM-Base [18] | 76.87 | 11.80 | 76.05 | 12.92 | 53.66 | 12.69 | 51.19 | 14.28 |
| REMONET (ours) | **83.12** | 09.52 | **82.39** | 10.88 | **62.91** | 12.47 | **60.28** | 13.10 |

Table 1: Performance (accuracies and F1 scores (%)) of different methods.

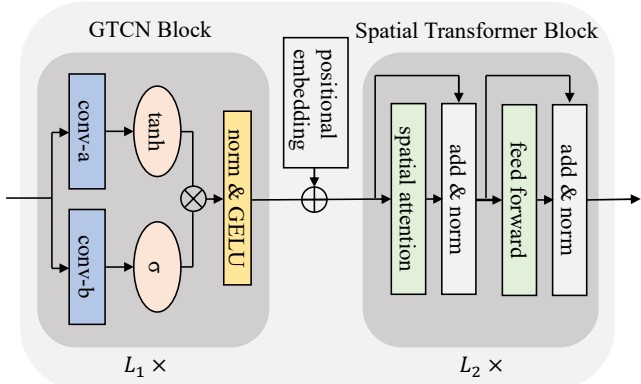

Figure 4: The hybrid architecture of the encoder. It is stacked by several gated TCN blocks and spatial Transformer blocks. The positional embeddings are added before the Transformer blocks.

sad, and fear). 15 subjects participated in the experiments 3 times in each dataset. Video clips are used as stimuli to elicit emotions. Each experiment contains 15 trials and 24 trials for SEED and SEED-IV, respectively. The 62-channel EEG signals are recorded with the international 10-20 system at the sampling rate of 1000 Hz.

For preprocessing, the EEG signals are first filtered with a band-pass of 0-75 Hz, and then downsampled to 200 Hz. DE features [6] that we utilize as the predicted target in pre-training are extracted from 5 frequency bands, i.e., $\delta$ (1-4 Hz), $\theta$ (4-8 Hz), $\alpha$ (8-14 Hz), $\beta$ (14-31 Hz), and $\gamma$ (31-50) Hz:

$$h(X) = -\int_X f(x) \log f(x) dx. \tag{15}$$

We follow the common settings of these two datasets to evaluate our framework. We use the first 9 trials as training data, and the last 6 trials as test data for SEED. For SEED-IV, the first 16 trials are used as training data and the remaining 8 trials are used as test data. The time window is 1 second for SEED and 4 seconds for SEED-IV. As the datasets are naturally noisy, the emotions at the beginning of videos are usually low-induced and the performance of evaluated methods might be inaccurate. Therefore, we remove the first 30 seconds of each trial in test data to get relatively clean test

data while the training data remain to be unchanged. All methods strictly follow the same conditions and are compared fairly. Our experiments are conducted under NVIDIA 2080 Ti with Python 3.10.9 and PyTorch 2.0.0.

## 4.2 Implementation Details

For the encoder, the number of GTCN blocks $L_1$ is 4, and the kernel sizes are [5, 5, 4, 2] for SEED and [10, 8, 5, 2] for SEED-IV. The number of spatial Transformer layers $L_2$ is 6 with 6 heads of multi-head attention. $L_h$ is set to 2 for the regressor while $L_g$ is 4 for the decoder. The feature predictor has $L_m = 3$ Transformer blocks and the embedding dimension is 16 for the projection head. The masking ratio $r$ is 0.5. The temperature $\tau$ for contrastive learning is set learnable. The batch size is 64 for both pre-training and fine-tuning. For pre-training, the model runs for 300 epochs with a learning rate of 0.0001 for both masked EEG feature prediction and contrastive learning and the AdamW [30] optimizer is applied with a weight decay of 0.05. The $\lambda$ is set to 2 and the momentum $\eta$ is 0.996. As for fine-tuning, the learning rate is tuned from {0.0003, 0.001, 0.003}, and the weight decay is selected from {0.0001, 0.01, 0.1}. The coefficient $\alpha$ is 1 while $\beta$ is tuned from 0.2 or 0.1. It should be noted that the GTCN blocks are frozen after pre-training since we observe performance decrease if they are tuned during fine-tuning. We guess this is because the large-scale pre-training endows the model with a powerful temporal processing capability. Fine-tuning with less data may do harm to the learned temporal feature extractor.

## 4.3 Comparison with Baselines

To compare REMONET with the existing methods, we consider six baselines. This includes several CNN-based methods: shallow convolutional networks (SCN), EEGNet, Filter-Bank Convolutional Network (FBCNet) which employs a multi-view representation followed by spatial filtering, and TSception which is a multi-scale convolutional neural network consisting of dynamic temporal, asymmetric spatial, and high-level fusion layers. There are also self-supervised baseline methods named Bert-inspired Neural Data Representations (BENDR), which adapts wav2vec 2.0 [1] in speech recognition to EEG with contrastive learning, and Large Brain Model (LaBraM), which is pre-trained on 2,500 hours EEG data through predicting masked neural codes [18]. Those methods are all end-to-end. To compare the performance with non-end-to-end

| #Channel | LaBraM-Base | | | | TSception | | | | REMoNET | | | |
|---|---|---|---|---|---|---|---|---|---|---|---|---|
| | ACC | STD | F1 | STD | ACC | STD | F1 | STD | ACC | STD | F1 | STD |
| 10 | 60.06 | 09.24 | 59.23 | 09.58 | 46.97 | 09.24 | 42.89 | 11.87 | **61.72** | 10.23 | **61.15** | 10.47 |
| 20 | 67.39 | 10.91 | 66.78 | 11.34 | 58.68 | 11.33 | 57.35 | 12.32 | **69.44** | 10.91 | **68.85** | 11.56 |
| 30 | 71.30 | 11.23 | 70.74 | 11.81 | 66.00 | 11.20 | 65.15 | 11.68 | **73.82** | 10.86 | **73.16** | 11.72 |
| 40 | 73.83 | 11.20 | 73.24 | 11.91 | 69.94 | 11.49 | 68.84 | 12.30 | **77.03** | 10.59 | **76.35** | 11.65 |
| 50 | 75.74 | 11.23 | 75.11 | 12.06 | 71.97 | 11.78 | 70.45 | 13.17 | **79.61** | 10.23 | **78.86** | 11.38 |

**Table 2: Results (accuracies and F1 scores (%)) of different methods for the simulation of corrupted EEG on the SEED dataset.**

approaches, we replicate the Masked Autoencoder using DE features. It worth noted that we don't compare with other existing non-end-to-end methods based on smoothed hand-crafted features owing to unfairness. The performance comparison results are listed in Table 1.

It can be observed that the proposed REMoNET outperforms all the other baseline approaches dramatically. This demonstrates the effectiveness of the self-supervised pre-training and robust fine-tuning. Specifically, our REMoNET achieves an average accuracy of 83.05% and a standard deviation of 9.6% on the SEED dataset, while no other method exceeds 80% in accuracy. Similarly, our framework surpasses other methods by a large margin with an accuracy of 62.91% and a standard deviation of 12.47% on the SEED-IV dataset while other methods are struggling with an accuracy of around 50%. Interestingly, the performance of MAE using unsmoothed DE features is comparable to that of other end-to-end approaches. Nevertheless, its standard deviation is much lower.

The videos of SEED and SEED-IV are determined by selecting videos with the highest score from the same stimuli pool, and for each emotion, SEED-IV has 8 videos while SEED has only 5 videos in a session, which means the videos of SEED-IV are considered less effective than SEED (since the mean score is lower). This fact indicates that SEED-IV is relatively noisier than SEED. For REMoNET, the performance gain on SEED-IV is 3.58%, which is higher than that on SEED (1.25%), demonstrating that REMoNET is more robust on datasets with noisier labels. This observation also highlights the flexibility of REMoNET as it can handle different noise levels without any prior knowledge.

## 4.4 Results of Corrupted EEG

As REMoNET is pre-trained by masked EEG channel modeling, it might have great potential in coping with corrupted EEG signals. We conduct experiments to simulate scenarios where some channels of EEG signals are randomly impaired or missing on the SEED dataset. In this case, the model is tested with different numbers of sound EEG channels (10, 20, 30, 40, and 50). We randomly sample the specified number of channels as the sound channels 50 times and compute the average test accuracy to make the results persuasive. To this end, all methods are trained with random corruption of EEG to obtain a more robust model. The experimental results are exhibited in Table 2. We compare our approach with the best baselines LaBraM-Base and TSception. It can be found that our model achieves the best accuracies in all cases. When the number of sound EEG channels becomes small, our method has a greater advantage

| DWR | AWR | SEED | SEED-IV |
|---|---|---|---|
| ✗ | ✗ | 81.80/09.88 | 59.33/**11.90** |
| ✗ | ✓ | 82.25/09.83 | 61.66/12.27 |
| ✓ | ✗ | 82.51/09.82 | 62.05/12.85 |
| ✓ | ✓ | **83.05/09.60** | **62.91**/12.47 |

**Table 3: Ablation study on multi-regularized fine-tuning (average accuracies and standard deviations (%)) of REMoNET. DWR and AWR denote distance-wise regularization and angle-wise regularization, respectively.**

| Method | SEED | | SEED-IV | |
|---|---|---|---|---|
| | ACC | F1 | ACC | F1 |
| Variant 1 | 80.95/10.17 | 80.09/**10.17** | 56.63/13.45 | 54.53/14.33 |
| Variant 2 | 81.80/09.88 | 81.08/10.94 | 59.33/11.90 | 59.15/**12.42** |
| Variant 3 | 82.68/09.71 | 82.00/10.86 | 61.24/14.17 | 59.34/13.92 |
| Variant 4 | 82.65/09.79 | 81.98/10.91 | 60.15/**11.42** | 58.90/13.29 |
| Ours | **83.12/09.52** | **82.39**/10.88 | **62.91**/12.47 | **60.28**/13.10 |

**Table 4: Ablation study on different variants of REMoNET (accuracies and F1 scores (%)) of different methods.**

over other methods. Even in extreme cases where only 10 channels are available, the proposed REMoNET still has an accuracy of 61.72% while the accuracies of other methods are obviously lower, especially TSception of 47%. In conclusion, these results demonstrate the robustness of REMoNET when dealing with corrupted EEG signals.

## 4.5 Ablation Study

To verify the effectiveness of multi-regularized co-learning, i.e., distance-wise regularization (DWR) and angle-wise regularization (AWR) in the fine-tuning stage, we perform the ablation study on both SEED and SEED-IV. Experiments with no DWR and no AWR, single DWR, single AWR, and both DWR and AWR are presented in Table 3. The accuracy decreases when using single regularization compared with both regularization terms. In addition, the results using single DWR are slightly better than those using single AWR, which indicates that DWR is more effective than AWR to some extent. Note that the performance of single AWR is improved as well compared to fine-tuning without DWR and AWR , demonstrating that AWR also plays an important role in dealing with noisy labels.

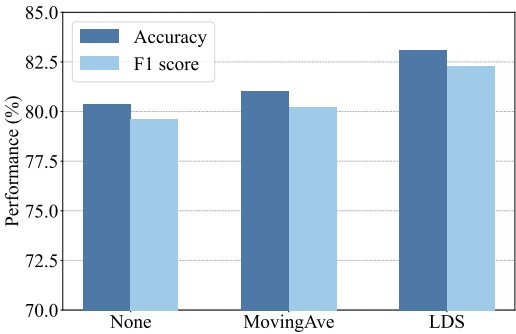

**Figure 5: The accuracies and F1 scores of different smoothing methods for the target features in the pre-training stage on the SEED dataset.**

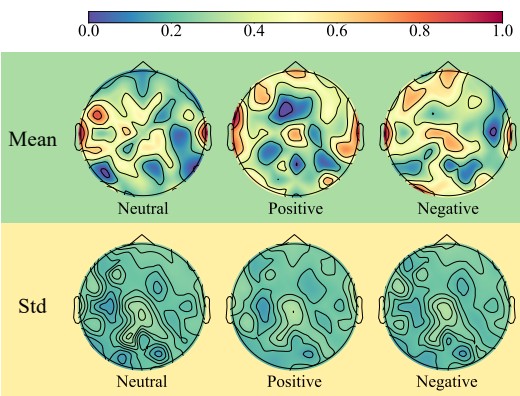

**Figure 6: Visualization of the attention weights on the SEED dataset. The redder the color, the larger the weight.**

Furthermore, we study four different variants of RᴇᴍᴏNᴇᴛ:

- Variant 1: Fine-tuning the model pre-trained by ECL in RᴇᴍᴏNᴇᴛ without any other optimization.
- Variant 2: Fine-tuning the model pre-trained by MCM-TST in RᴇᴍᴏNᴇᴛ without any other optimization.
- Variant 3: Discarding the gating mechanism in GTCN.
- Variant 4: Fine-tuning two different pre-trained models jointly with co-regularization in JoCoR [41], which is a robust training scheme for noise reduction.

The results are shown in Table 4. It can be induced that masked EEG feature prediction (Variant 2) has better performance than contrastive learning (Variant 1). This observation illustrates that predicting semantic features from unmasked EEG signals can be more effective in capturing emotion-related information. However, both these two fine-tuned models are superior to other baselines in Table 1, which further proves the superiority of self-supervised pre-training. Compared with Variant 2, RᴇᴍᴏNᴇᴛ exceeds 1.25% accuracy on SEED and 3.58% accuracy on SEED-IV. In addition, Variant 3 showcases the effectiveness of the gated mechanism of GTCN. Compared with Variant 4, RᴇᴍᴏNᴇᴛ also shows its improvements, indicating the superiority of utilizing self-supervision in handling noisy labels.

### 4.6 Impact of Smoothing Target Features

We further investigate the impact of smoothing on target features in the pre-training stage. Although feature smoothing is not suitable for real-time application scenarios, it is feasible to utilize it during the model's offline pre-training for learning better representation. We consider two widely used feature smoothing algorithms: moving average and linear dynamic system (LDS). For a specific feature, moving average simply takes the weighted sum of the features in a small time window while LDS has the observation of the whole trial. The comparison is illustrated in Figure 5. Firstly, features with smoothing perform better than non-smoothing. This is consistent with previous studies that directly used smoothed features to recognize emotions, indicating that smoothed features are also helpful for self-supervised pre-training. Additionally, LDS outperforms moving average, which demonstrates that it is effective to take the time dependency of the whole trial into consideration.

### 4.7 Visualization

Since the self-attention mechanism that we adopt in the spatial Transformer block can automatically assign dynamic weights on EEG channels depending on the input data, it is capable to explore the critical channels. We visualize the attention weights (mean and standard deviation) of neutral, positive, and negative emotions in RᴇᴍᴏNᴇᴛ to investigate the critical EEG channels for emotion recognition, as depicted in Figure 6. The attention weights of all data and all subjects are first min-max normalized and then calculated by averaging or calculating the standard deviation of the attention weights through all heads of the last Transformer blocks between all channels and the class tokens. It can be observed that the attention weights in the lateral temporal areas have high values for all emotions, indicating that the lateral temporal areas are critical for emotion recognition. This finding is mostly consistent with previous work [48] using the DE features. Moreover, the weights in the frontal and parietal lobes of positive and negative emotions are obviously higher than the neutral emotion. Furthermore, the occipital area of the negative emotion has higher attention weights. The low standard deviation among all brain regions proves the stability of our observation.

### 5 CONCLUSION

In this paper, we have introduced a comprehensive pipeline named RᴇᴍᴏNᴇᴛ aiming at reducing emotional label noise through multi-regularized self-supervision. Such self-supervision methods also enable a comprehensive interpretation of the multifaceted characteristics of EEG signals, facilitating the identification of the most relevant factors for emotion recognition. Our proposed method exhibits significant potential in handling corrupted EEG signals and demonstrates proficiency in processing varying numbers of EEG channels compared to existing methods. Additionally, we utilized an attention mechanism to explore critical EEG channels for emotion recognition. The experiments conducted on corrupted EEG data and the identification of critical channels provide insights into potentially reducing the number of EEG electrodes. We hope this preliminary work empowers the feasibility of human emotion recognition and intricate EEG signals.

 

## ACKNOWLEDGMENTS

This work was supported in part by grants from STI 2030-Major Projects+2022ZD0208500, Shanghai Municipal Science and Technology Major Project (Grant No. 2021SHZDZX), Medical-Engineering Interdisciplinary Research Foundation of Shanghai Jiao Tong University "Jiao Tong Star" Program (YG2023ZD25 and YG2024ZD25), Shanghai Pilot Program for Basic Research - Shanghai Jiao Tong University (No. 21TQ1400203), and GuangCi Professorship Program of RuiJin Hospital Shanghai Jiao Tong University School of Medicine.

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
