# OpenReview forum: "REmoNet: Reducing Emotional Label Noise via Multi-regularized Self-supervision"
_acmmm.org/ACMMM/2024/Conference — MM2024 Poster_

### Official Review · Reviewer_9cLa · 2024-05-13

**Rating:** 5
**Confidence:** 3

**Summary:**

The paper presents a comprehensive pipeline named REmoNet designed to enhance emotion recognition from electroencephalogram (EEG) signals. The key focus of the paper is to address the challenges associated with emotional label noise and the complexities of capturing the temporal-spatial-spectral characteristics of EEG signals, especially in the presence of low signal-to-noise ratios (SNR).

**Strengths:**

1. The paper introduces a novel self-supervised learning technique combined with multi-regularized co-learning to reduce emotional label noise, which is a significant issue in EEG-based emotion recognition.
2. The authors conducted experiments on two public datasets, SEED and SEED-IV, demonstrating that REmoNet outperforms existing state-of-the-art methods in handling raw EEG signals and noisy emotional labels.
3. The use of an attention mechanism to explore critical EEG channels provides insights that could potentially reduce the number of EEG electrodes needed, simplifying the process.
4. The paper shows that REmoNet is robust against corrupted EEG signals and can handle varying numbers of EEG channels, which is a significant advantage in real-world applications where data integrity might be compromised.

**Limitations:**

1.  The method is tailored to address label noise; however, the effectiveness of the approach might be diminished in scenarios where label noise is not the primary challenge.
2. The paper does not discuss the computational complexity or resource requirements of the proposed method, which could be a concern for its practical implementation, especially in resource-constrained environments.

**Suitability:**

3

---

### Official Review · Reviewer_CyHX · 2024-05-24

**Rating:** 3
**Confidence:** 4

**Summary:**

This paper focuses on the challenges of label noise and the difficulty of extracting temporal-spatial features in EEG-based emotion recognition. This is paper propose REmoNet, a comprehensive pipeline utilizing self-supervised techniques and multi-regularized co-learning. This approach includes masked channel modeling, emotion contrastive learning, and fine-tuning with multi-regularized co-learning to improve the understanding and extraction of emotion-relevant EEG representations.

**Strengths:**

1.The pre-training method described in this article for feature extraction from EEG signals is worth studying.

**Limitations:**

1.The method proposed in this paper aims to address the issue of label noise. From the Introduction and experimental sections, we find that the paper assumes that the SEED dataset is noisy, with the noise being present in the first 30 seconds of the signal. This assumption is inconsistent with other papers addressing label noise, which are cited in the study. Most of these papers assume that the public datasets are clean and artificially add noise.

2.We found a paper that also addresses label noise in EEG-based emotion recognition. The paper introduces artificial noise into an existing dataset.However, this paper does not cite this relevant work. Can you compare with the method in that paper?
“Multi-channel EEG-based emotion recognition in the presence of noisy labels.”

3.The note of Figure 3.3 indicates that parameters in the model are frozen or partially frozen, but Section 3.3 does not give an explanation about it.

4.The method proposed in this paper is an application of existing method in EEG-based emotion recognition. Its innovation is not significant.
5.In the ablation study, this paper compares JoCoR [35]. We don’t understand why authorr  compared this method in the ablation study. Additionally, reference [35] was published in 2020, while a similar work[31] was proposed in 2021. If the comparison is needed, why not compare with the method in reference [31]?

**Suitability:**

2

---

### Official Review · Reviewer_uJWb · 2024-05-24

**Rating:** 3
**Confidence:** 3

**Summary:**

This work focuses on EEG-based emotion recognition, particularly using self-supervised methods. The proposed framework consists of three main parts:
1. A masking strategy based on temporal-spectral transformation.
2. Traditional contrastive learning.
3. A co-teaching fine-tuning stage to align the feature space of the classifier and the pre-trained projector.

The framework was evaluated on the SEED and SEED-IV datasets. Although the framework is complex with numerous design elements, making it interesting and inspiring, it lacks an ablation study for the first two parts to demonstrate the necessity of their components or configurations. Some components of the architecture seem simplifiable while still achieving the same goal, and some configurations are not convincing without comparison. The absence of such an ablation study significantly impacts the completeness and reliability of this work.

**Strengths:**

- The integration of three techniques—masked strategy, contrastive learning, and multi-regularized co-learning—into an EEG-based emotion recognition framework effectively addresses noisy labels.
- The proposed framework was validated on two well-known public datasets, outperforming some SOTA methods.

**Limitations:**

#### **Methodology and Experiments**

- Necessity of the Regressor \(h\): The authors apply convolutions only on the time dimension through the encoder \(f\) due to masked channels. Therefore, an additional network, Regressor \(h\), is added to infer the embeddings of the masked channels. However, using a single temporal-spatial network might directly reconstruct both temporal and spatial features simultaneously, as demonstrated in other studies (e.g., Chien et al., NeurIPS 2022).

- The use of a transformation pipeline to build the masked autoencoder raises questions. If emotional dynamics are reflected at higher frequency bands, directly using spectral features as inputs might be more effective. Previous studies (e.g., Li et al., ACM MM 2023) suggest training a specific encoder for spectral characteristics. The method used by the authors seems more like using a complex network as an alternative to traditional spectral feature extraction. Additionally, did the authors encounter the same issues as the LabraM model, which encodes EEG signals into FFT amplitude and phase due to reconstruction challenges?

- Using samples from different subjects evoked by the same video clips as negative pairs might affect the generalizability and transferability of the model across subjects.

- The absence of an ablation study to demonstrate the effectiveness and necessity of the main components is a significant limitation. For instance, the claim that gated 1-D CNN should be better than 1-D CNN lacks experimental evidence, as does the comparison between using an EEG-to-spectral features masked autoencoder and a spectral reconstruction masked autoencoder.

- In figure 6, the average normalized attention weights across subjects might not express the causal relationship between emotional evocation and brain region activation. Considering inter-subject variability, adding the standard deviation of non-normalized individual attention maps would provide better insights into the consistency across individuals.

- Given the complex architecture and some unclear design details, providing open-source code would greatly benefit the community.

#### **Writing and Logic**

- The motivation for using EEG for emotion recognition should be revised. For example, the introduction mentions, "Various modalities have been utilized, including non-physiological signals like facial expression [20] and speech [32], physiological signals like electroencephalography (EEG) [14] and electrocardiogram [13]. Among the diverse modalities, EEG has garnered significant attention due to its non-invasive nature and high temporal resolution." However, modalities like facial expression, speech, and ECG are also non-invasive, and both speech and ECG have high temporal resolution.

#### **References**

- The references could be improved by selecting those that are more relevant to the context and have a significant impact on the field. For instance, regarding EEG-based emotion recognition methods using deep neural networks, the authors cited two works [16][38] that focus more on foundational EEG models pre-trained on various large-scale EEG datasets, rather than on emotion recognition, which is the main topic of the authors' work.

**Suitability:**

3

---

### Meta-Review · Area_Chair_cUcs · 2024-07-11

**Recommendation:** Accept (Poster)
**Confidence:** 4

**Metareview:**

This is an interesting paper and was of interest to all reviewers. The rebuttal was clear and concise and addressed many of the concerns of the reviewers and has resulted in improved scores.

For the final version of the paper, all of the comments as addressed in the rebuttal should be included.